# Use of E-Health in Dutch General Practice during the COVID-19 Pandemic

**DOI:** 10.3390/ijerph182312479

**Published:** 2021-11-26

**Authors:** Jelle Keuper, Ronald Batenburg, Robert Verheij, Lilian van Tuyl

**Affiliations:** 1Netherlands Institute for Health Services Research (NIVEL), 3513 CR Utrecht, The Netherlands; r.batenburg@nivel.nl (R.B.); r.verheij@nivel.nl (R.V.); l.vantuyl@nivel.nl (L.v.T.); 2Tranzo, Tilburg University, 5037 DB Tilburg, The Netherlands; 3Department of Sociology, Radboud University Nijmegen, 6525 XZ Nijmegen, The Netherlands

**Keywords:** general practice, e-health, COVID-19

## Abstract

The COVID-19 pandemic has forced general practices to search for possibilities to provide healthcare remotely (e.g., e-health). In this study, the impact of the pandemic on the use of e-health in general practices in the Netherlands was investigated. In addition, the intention of practices to continue using e-health more intensively and differences in the use of e-health between practice types were investigated. For this purpose, web surveys were sent to general practices in April and July 2020. Descriptive data analysis was performed and differences in the use of e-health between practice types were tested using one-way ANOVA. Response rates were 34% (*n* = 1433) in April and 17% (*n* = 719) in July. The pandemic invoked an increased use of several (new) e-health applications. A minority of practices indicated the intention to maintain this increased use. In addition, small differences in the use of e-health between the different practice types were found. This study showed that although there was an increased uptake of e-health in Dutch general practice during the COVID-19 pandemic, only a minority of practices intends to maintain this increased use in the future. This may point towards a temporary uptake of digital healthcare delivery rather than accelerated implementation of digital processes.

## 1. Introduction

The coronavirus disease 2019 (COVID-19) pandemic has impacted healthcare delivery worldwide [1]. With social distancing as one of the main measures taken by governments to control the spread of the COVID-19 virus, healthcare providers were forced to become creative in providing care [2]. Traditional physical consultations needed to be reduced to a minimum during the pandemic, creating opportunities for innovative forms of healthcare delivery.

This need to change can be clearly seen in the organization of Dutch primary care during the COVID-19 pandemic. In the Netherlands, general practices operate as the first point of contact for citizens having health problems and questions about their health [3]. The practices are small scale private enterprises, generally owned by one or several general practice owners, and funded via a system of fee for service and capitation fees [4]. Access to medical specialist care is only possible after a referral of the general practitioner (GP). General practice care is free of charge in the Netherlands and almost every citizen is registered as a patient at one general practice. At the beginning of the pandemic in the Netherlands, in March 2020, general practices were flooded with telephone calls from patients and needed to deal with waiting rooms too small to allow for sufficient distance between patients. In addition, the use of general practice care declined during the initial phase of the pandemic [5]. These trends triggered more use of remote healthcare options (e.g., e-health), such as e-consultations and video consultations, which provided solutions to continue healthcare delivery to patients [6].

Since 2013, the availability and use of e-health by GPs has been monitored annually through the Dutch eHealth-monitor [7]. Based on this monitor, we defined e-health in this study as: *the use of new information and communication technologies, in particular internet technology, to improve health and support or improve healthcare*. This eHealth-monitor demonstrates a gradual increase of the use of e-health in Dutch general practice from 2013 until 2019. 2019, in particular, saw the adoption of online ordering of repeat prescriptions (82% of the GPs), teleconsultations (73%), and e-consultations (68%). Telemonitoring, however, was used by only 18% of the GPs, and video consultations were rarely provided at all. In the Dutch eHealth-monitor and in our study, a video consultation is considered as a real-time visual and audio (digital) contact moment between the patient and healthcare provider, while an e-consultation facilitates asynchronous written (digital) contact between the patient and the GP. A teleconsultation is considered as a (digital) contact moment between the GP and a medical specialist. Telemonitoring is regarded as digital self-monitoring of health data by patients. Compared to 27 other European countries, the number of general practices that has adopted e-health technologies in the Netherlands appears relatively high [8]. Consultations with other healthcare practitioners (teleconsultations) were adopted on average by 21% of the almost 6000 consulted GPs, consultations with patients (e-consultations) were adopted on average by 12%, and telemonitoring on average by 5%.

While it is already clear from other studies that the COVID-19 pandemic has greatly influenced the use of e-health technology by general practices, it remains unknown which specific e-health applications were impacted most by the pandemic, if this use will increase only temporarily or structurally, and what variation exists in the uptake by type of e-health and type of general practices. In this paper, we consequently describe (1) the adoption and (2) use of several e-health tools in Dutch general practice during the initial phase of the pandemic, and (3) the intention of practices to continue using e-health applications in the near future (post-COVID-19). Variation in the adoption, use, and intention across practices (4) is also described, aiming to explore if these variations are related to characteristics of the e-health application and/or characteristics of the general practices. Herewith, we tested the expectation that the intention of practices to maintain the use of e-health in the future differs with practice size (in terms of GP workforce in general practices) and the type of e-health application.

## 2. Materials and Methods

### 2.1. Sample and Data Collection

In the Netherlands, there are nearly 5000 general practices. In April 2020, 4168 practices were approached by a personal e-mail invitation to participate in the web surveys. In July 2020, 4218 practices were contacted to participate in this study. These two moments were deliberately chosen to investigate the use of e-health at two different points of time during the pandemic. At the end of April, the Netherlands was in an “intelligent” (partial) lockdown as the number of infections and hospitalizations were peaking. In July 2020, the first “intelligent” (partial) lockdown had ended, only mild governmental measures applied, and the number of infections and hospitalizations were low. Contact details of all Dutch general practices, such as e-mail addresses, are collected and monitored continuously through NIVEL’s Healthcare Professionals Registries [9]. Each general practice received a personalized weblink to their web survey, which was completed by one respondent of the practice. To increase attention to these web surveys, we sent out personal e-mail reminders and additional reminders using the social media accounts of the research institute. The response data was collected and processed anonymously, and afterwards provided securely to the researchers. Information about the name of the general practice was not available for the researchers.

### 2.2. Web Survey

The two surveys were composed by researchers and were tested by GPs using Survalyzer software. Questions were posed about the impact of the COVID-19 pandemic on the practice organization, patients, staff, and the healthcare delivery process. Both surveys also included questions regarding the use of e-health and the intention to continue using e-health in the future (see Appendix A). The Theory of Planned Behavior, with a direct measure of intention to use e-health in the future, was used as inspiration in setting up the survey questions [10]. The use of e-health was measured by five pre-specified and commonly used e-health applications in Dutch general practice, namely: e-consultations, video consultations, online ordering of repeat prescriptions, teleconsultations, and telemonitoring. In addition, respondents could specify which other e-health applications they were using in an open response field.

### 2.3. General Practice Type Definition

Background data about the general practices were collected by the Healthcare Professionals Registries through their websites before sending out the web survey. These data were anonymously provided to the researchers. This concerns the location(s) of the practice and the number of GPs actively practicing. With only one practicing GP, the practice type was coded as a solo practice, two practicing GPs as a duo practice, and with three or more GPs working in a practice the practice type was coded as a group practice.

### 2.4. Data Analysis

Descriptive data analysis was performed using IBM SPSS Statistics v27 and Microsoft Excel v2010 software. Unknown background data about the general practices was treated as missing data. If applicable, the open responses of the category ‘other, namely’ (see Appendix A) were recoded and added up to the response of the pre-specified e-health applications. Differences in use of the five specified e-health tools between the three practice types were tested by performing one-way ANOVA. These were regarded as significant if the two-sided *p*-value was 0.05 or lower.

## 3. Results

### 3.1. Sample Characteristics

In April 2020, the response rate was 34% (*n* = 1433), based on fully completed web surveys. In July 2020, the response rate was 17% (*n* = 719). The majority of surveys (85% and 76% respectively) were completed by general practice owners. A smaller percentage (6% and 12% respectively) were completed by practice managers (see Appendix A). The distribution of solo, duo, and group practices in our study was comparable to the distribution in the total population (see Appendix A).

### 3.2. Use of E-Health

In total, 1083 practices (75%) of the responding practices in April answered that they started using (new) e-health applications or that they intensified the use of existing e-health applications due to the start of the COVID-19 pandemic. A relatively high proportion of practices (64%) reported the first time use of video consultations and 8% revealed intensified use of this e-health application due to the COVID-19 pandemic. E-consultations, online ordering of repeat prescriptions, and teleconsultations were already used by the majority of the general practices before the pandemic. A small proportion reported first time use of these applications, whilst a high proportion of general practices revealed intensified use (52%, 68%, and 73%, respectively). Use of telemonitoring remained low during the pandemic: 5% used this for the first time and 13% of the practices intensified its use (Figure 1). Reported first time use and intensified use of other e-health applications (not specified in Figure 1) were mainly related to teledermatology, e-mental health programs, chat programs, online access for patients to their electronic health record, online patient portals, and online video meetings with other healthcare professionals.

In July, online ordering of repeat prescriptions and e-consultations were the most commonly adopted e-health applications (Figure 2). They were used by 87% and 84% of the general practices, respectively. About two thirds of all practices (66%) used teleconsultations. About half of the practices (52%) were applying video consultations, and 12% mentioned the use of telemonitoring. Other e-health applications were indicated by 5% of the practices and were similar to those indicated in April, but also included e-appointments. Results of the second survey showed that almost all (98%) general practices reported the use of at least one e-health application. More than two-thirds (70%) of the practices used at least three e-health applications.

A small proportion of practices (2%) that indicated no use of e-health could clarify their answer in the survey and indicated that they had contact with patients by telephone, or continued consulting their patients in person, strictly adhering to the governmental physical measures and other precautions. One practice started using e-health applications during the pandemic, but quickly ceased its use: “We used (video) consultation and e-consultations during the pandemic, but now we are acting ‘normal’ again. We would like to see patients in person if that is possible, we do not see the added value of video consultation, on the contrary!” Another practice indicated: “Now we operate as before. Video consultation has been used, however now the consultations are face-to-face again. Previously we have done a lot [of the patient contacts] by telephone. The elderly population is not always able to use the computer, but they are able to use the telephone”.

### 3.3. Intention to Maintain Use of E-Health

Figure 3 presents the intention of general practices to continue e-health applications after the pandemic. In April, 28% reported the intention to use video consultations more intensively after the pandemic. For e-consultations, teleconsultations, online ordering of repeat prescriptions, and telemonitoring, these percentages were 26%, 17%, 14%, and 7%, respectively.

In July, these proportions were somewhat higher for e-consultations, teleconsultations, and online ordering of repeat prescriptions, and 32% of the practices indicated that they intended to continue using e-consultations more intensively after the pandemic. For online ordering of repeat prescriptions and teleconsultations, these percentages were both 23%. The proportions for video consultations (20%) and telemonitoring (5%) were somewhat lower compared to April.

### 3.4. Practice Type Differences

Table 1 shows that solo practices indicated more often that they were using e-consultations and teleconsultations for the first time compared to duo and group practices in April. Group practices indicated first time use of video consultations more often compared to the other two practice types. The first time use of online ordering of repeat prescriptions, telemonitoring, and other e-health applications was almost the same among the three types of practices. ANOVA tests indicated that only the differences in first time use of teleconsultations were significant between the three types of practices.

Based on the results from July, similar (small) differences between the three practice types were found in the use of video consultations, online ordering of repeat prescriptions, teleconsultations, and other e-health applications. A similar result to the April survey is that video consultations were used more often in group practices compared to smaller practice types. Telemonitoring was used most by duo practices. Furthermore, almost all duo and group practices indicated that they were using at least one e-health application in their practice (Table 2).

Subsequently, we also analyzed the differences between the three general practice types with regard to their intensified use of the five specified e-health applications, both in April and July. In April, we found that group practices more often indicated intensified use of e-consultations and teleconsultations compared to solo practices and duo practices. The intensified use of online ordering of repeat prescriptions, video consultations, telemonitoring, and other e-health applications were almost the same among the different types of practices. ANOVA tests indicated that only the differences in intensified use of e-consultations were significant between the three types of practices in April (see Appendix A). No significant differences were found in the results of July. However, intensified use of e-consultations, video consultations, teleconsultations, and telemonitoring was slightly lower in solo practices compared to duo and group practices (see Appendix A).

Finally, we explored if practices differed in their intention to continue using e-health more intensively after the COVID-19 pandemic. In April, solo practices indicated more often that they intend to maintain the intensified use of telemonitoring after the COVID-19 pandemic, compared to duo practices and group practices. In addition, group practices indicated more often that they intend to continue using video consultations more intensively after the pandemic, compared to solo and duo practices. For the other e-health applications, answers did not differ considerably among the different types of practices. ANOVA tests indicated that the differences in the intention to continue the intensified use of video consultations and telemonitoring were significant between the three types of practices (see Appendix A). In July, a significant difference was only found for online ordering of repeat prescriptions. Solo and duo practices more often intended to continue using this application more intensively after the COVID-19 pandemic, compared to group practices. The intention to maintain the intensified use of telemonitoring and other e-health applications was higher among solo practices compared to duo and group practices (see Appendix A).

## 4. Discussion

This study investigated the impact of the COVID-19 pandemic on the uptake and use of e-health in Dutch general practices and their intention to continue the use of e-health after the pandemic. Since the start of the pandemic, the use of e-health has rapidly increased. We found that the use of video consultations has increased most in April 2020 compared to 2019, in terms of first time use as well as the intention to continue the use of this application. This could indicate that video consultations are relatively easy and rapid to implement, most probably because the video communicating tools had already existed for some years in the Netherlands. As we know from the adoption literature, an important factor is that people are familiar with (new) technologies and clearly see the advantages of these [11,12,13]. In addition, the digital infrastructure is of the highest standard and consequently good enough to enable video consultations all over the country. In contrast, the use of e-consultations and online ordering of repeat prescriptions has only slightly increased because of the pandemic, whilst the use of telemonitoring seemed unchanged compared to the pre-pandemic period. This may be explained by the fact that telemonitoring applications consist of more functionalities, requiring more investment (time investment to train patients and/or personnel) and workflow changes, as they are mainly not integrated into the current digital infrastructure of the general practices. Apart from the technical issues, the question might also be whether general practices are equipped to fulfill a monitoring role. In addition, it might also be of less urgency and critical added value as the COVID-19 pandemic forced GPs to focus on patients with urgent complaints, diseases, and illnesses relative to focusing on the prevention of illnesses.

An increase in the use of e-health during the COVID-19 pandemic has also been reported by some other high-income countries. In Norway and Australia, for example, also an extreme increase in the use of video consultations was found in primary care during the initial phase of the pandemic [14,15,16]. In the United States, a rise in the use of telehealth was reported because of the pandemic [17]. In the region of Catalonia (Spain), 70% of previous face-to-face consultations took place online because of the pandemic. In addition, Germany has experienced an enormous increase in the number of teleconsultations, with growth rates in use of over 1000% [18]. Additionally, in low- and middle-income countries, such as India, the use of e-health has risen [19]. However, uptake of e-health in such countries might be more difficult compared to high-income countries, because they generally have lower investments and limited internet connections [20]. In contrast to these studies, our study provides insight into the intention of GPs to continue the use of these digital tools in their practice in the future. This is an important attribute, as the adoption of digital tools in healthcare delivery is generally seen as one of the solutions of the problems healthcare organizations face, including the ever rising costs, perceived (high) workloads, and shortages in healthcare workforce [21].

However, we observed that the majority of general practices do not intend to continue their intensified use of e-health after the pandemic. This may be due to the fact that the intensified use was necessitated by the pandemic in order to continue providing the required patient care. General practices were consequently compelled to quickly decide on implementing e-health services which could facilitate in providing the necessary care to their patients without the need to see them physically. Normally, implementing e-health is a long-term process, including several steps (e.g., preceding considerations, multidisciplinary assessment, and assessment of transferability) to ensure that the e-health application fits well with the organization of the general practice. Several frameworks and codes have been set up and research has been conducted to assist healthcare organizations in deliberately choosing the right e-health service and to evaluate its use [22,23,24,25,26]. Due to time pressure, general practices might not have considered these tools. The accompanying costs of e-health applications, which is an important economic aspect for general practices, may also be perceived as too high to sustain their usage [23]. Looking to other studies, physician perspectives about the use of e-health might also play an important role. Florea et al. found that GPs in Romania had an overall positive perception of the use of e-health. However, the time-consuming nature of teleconsultations, uncertainty in tele-decisions, and patients’ difficulties in using technology were seen as barriers for the rapid implementation of e-health applications [27]. Gomez et al. also investigated primary care physician perspectives of rapidly implemented e-health applications during the pandemic. Participants indicated that telemedicine improved patient access to care by providing greater convenience. At the same time, however, concerns were expressed that certain vulnerable patient groups were unable to use or did not possess the technology required to participate in telemedicine visits. In addition, the use of e-health was challenging when visits required a physical examination, such as in the case of spirometry tests for COPD patients. Further, respondents indicated that they were concerned about the loss of personal connections and touch, which can strengthen physician-patient relationships [28].

The analyses of e-health use that were performed in relation to practice characteristics showed small differences in the use of e-health applications. We limited our analyses to the differences between practices with one, two, and three or more actively practicing GPs, as a proxy for the practice type, therewith looking at the amount of resources and coordination. Overall, it appeared that group practices were using e-health applications more than solo- or duo-practices. This finding is in line with results from Australia, where GPs in larger practices had higher proportions of telehealth consultations compared to GPs from solo practices [29]. This result might be explained by the fact that resources in terms of workforce, money, and time for implementation of e-health applications are larger in group practices. In addition, group practices are likely to have more managerial and ICT staff with knowledge of implementing and using e-health, compared to solo or duo practices. Another factor could be that group practices have more personnel available, which allows more possibilities to learn to cope with new technologies. Several of these facilitating factors were investigated in a study by Bush et al. who inquired to information system managers in healthcare organizations about hindering and facilitating organizational characteristics in the process of implementing new information systems [22].

Looking at the differences in the use of e-health between the two distinct moments, we showed that the use of video consultations in July was much lower compared to the first time use and intensified use of this application together in April. In addition, the intention to continue the intensified use of video consultations in July was lower compared to April, whilst the intention to continue the intensified use of e-consultations, online ordering of repeat prescriptions and teleconsultations slightly increased in July compared to April. This could be explained by the fact that most general practices were already familiar with the use of these three applications, which were already embedded in their workflow. Video consultations were rapidly implemented by most practices, when part of the patients could not be seen face-to-face. In July, most general practices could physically see their patients again, which made the use of video consultations less needed. In addition, practices might prefer the use of face-to-face consultations.

Based on other literature we have found about the impact of the COVID-19 pandemic on the use of e-health in general practices, it can be concluded that our study is most probably the first longitudinal study to investigate the impact of the COVID-19 pandemic on the (first time and intensified) use and intention to continue using the five specified e-health applications more intensively in general practice after the pandemic. Our results provide evidence that e-health applications were more broadly used and built upon the existing usage, especially in duo and group practices. The intensified use of several e-health applications reflects a quest to provide remote care by rapidly implementing new and existing e-health applications during the COVID-19 pandemic. This may have an impact on the provision and quality of patient care, as patients would also be informed and advised to make more use of digital care, leading to less physical contacts between healthcare professionals and patients. However, there is also a group of professionals and patients who have difficulty using these new technologies, which might have an impact on the quality of care. The Dutch government could play an important role in facilitating the process of becoming familiar with e-health and to sustain the increased use of e-health, as it is generally seen as an important solution to overcome the problems (e.g., high workload and shortages in workforce) healthcare organizations face. However, from our results it is expected that the current intensified use of several e-health applications will slow down or even decrease after the pandemic.

Some limitations of our study should be pointed out. Firstly, the questions about the use of e-health and the intention to continue using it more intensively differed slightly between the web-surveys of April and July (see Appendix A). Consequently, it is difficult to compare these results properly. Furthermore, this study is based on a limited set of questions related to the use of e-health in general practices, as these questions were part of a larger survey about the impact of COVID-19 on the practice organization, patients, staff, and the healthcare delivery process. In addition, the response rate in July was lower compared to April, which might make the comparison between the two measurements debatable. The lower response rate in July might be explained by the fact that COVID-19 was less problematic at that moment and because it fell within a period of summer holidays. Looking at the practice sample representativeness, there were, however, only small differences between April and July. Further, the increased uptake of e-health might also be caused by factors other than the COVID-19 pandemic, such as policy regulations by the Ministry of Health and healthcare insurers in the last few years. In addition, Dutch healthcare insurers currently reimburse e-consultations and video consultations in the same way as face-to-face consultations, which has made it easier for practices to use e-health. However, despite these limitations, we believe that the presented data provide relevant insights into the impact of the COVID-19 pandemic on the adoption and use of e-health applications in general practices.

Future research could focus on how general practice staff and their patients experienced the (increased) use of e-health services during the COVID-19 pandemic, if they had technical problems with implementing these services, and how much time they spent using the several e-health services. In addition, it would be interesting to investigate staff organization changes in general practice related to the use of e-health. Furthermore, future research could investigate the reasons why the majority of the practices indicated that they did not want to continue the use of e-health more intensively in the future. Miner et al., for example, studied factors associated with clinicians’ intention to continue telemedicine services after the pandemic. They found that this intention was associated with a higher satisfaction by the clinician with the quality provided by telemedicine services, approval of the ease of performing a physical examination with these services, belief of the clinician that adaptability is an important element of being a clinician, and also with less preference for in-person work meetings over virtual meetings [30]. Another study highlighted five key requirements for the long-term sustainability of telehealth, namely the development of skilled workforce, the empowerment of patients, the reforming of funding, the improvement of the digital ecosystems, and integration of telehealth into routine care [31]. Similar research could also be executed in the context of Dutch general practice care in order to find out which factors still hinder the successful uptake of e-health. Further, a next step would be to investigate the impact of increased use of e-health on the quality of provided care, especially the impact of the rapidly implemented e-health applications. Consequently, surveys among the same practices are needed to investigate the use of e-health among patients and their perspectives on the use of e-health. This will provide multi-level data that can provide further insights into the causes and factors for e-health adoption and usage by general practices.

## 5. Conclusions

This study showed that Dutch general practices intensified the use of e-health during the COVID-19 pandemic. We found that 75% of the practices indicated the first time use of new e-health applications or the intensified use of already existing e-health applications, at the initial phase of the pandemic in April 2020. In particular, the use of video consultations was new for most of the general practices. Intensified use of e-consultations and online ordering of repeat prescriptions was also reported by a large majority of the practices. On the contrary, only 12% of the practices answered that they were using a form of telemonitoring in their practice. Further, a minority of the practices expected intensified use of the five pre-specified e-health applications in the future. In addition, some small differences in the use of e-health between the different types of practices were found. Overall, use of e-health is slightly higher among group practices, all of which indicated the use of at least one e-health application. These findings may point towards a temporary uptake of digital healthcare delivery, caused by the COVID-19 pandemic, rather than an accelerated sustainable implementation of digital processes.

## Figures and Tables

**Figure 1 ijerph-18-12479-f001:**
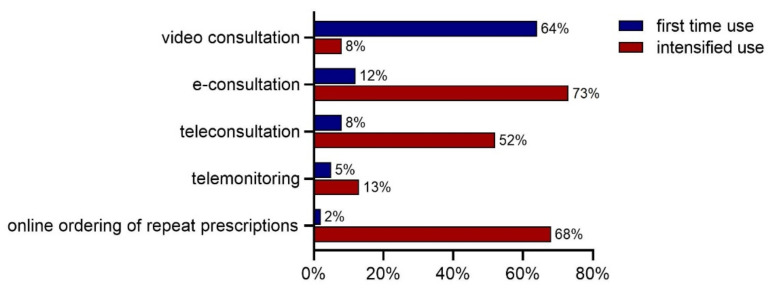
First time use and intensified use of (new) e-health applications, *n* = 1083 (April 2020).

**Figure 2 ijerph-18-12479-f002:**
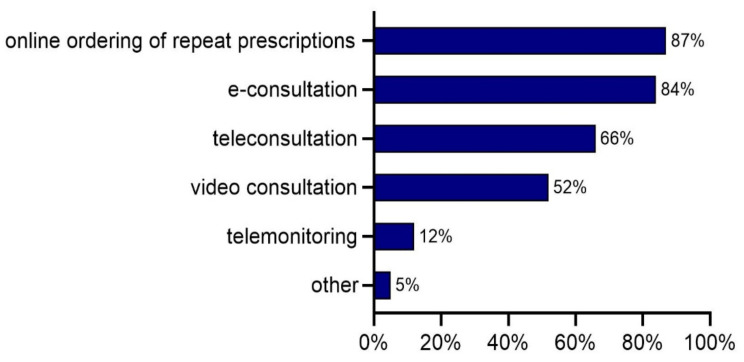
Percentage of practices that used a specific e-health application, *n* = 732 (July 2020).

**Figure 3 ijerph-18-12479-f003:**
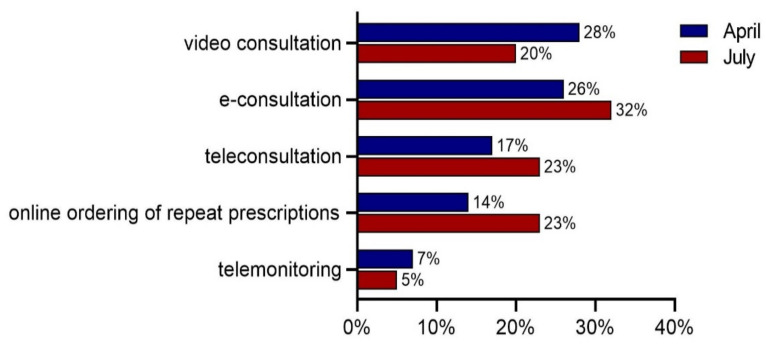
Percentage of practices that expected continuation of intensified use of specific e-health applications after the pandemic, *n* = 1083 (April 2020) and *n* = 718 (July 2020).

**Table 1 ijerph-18-12479-t001:** Percentage of practices using e-health applications for the first time because of the COVID-19 pandemic, specified by e-health application (mean ± standard deviation), *n* = 1074 (April 2020).

E-Health Application	Solo Practice (*n* = 211)	Duo Practice (*n* = 480)	Group Practice (*n* = 383)	*p*-Value
E-consultation	15% ± 0.355	12% ± 0.326	9% ± 0.289	0.114
Online ordering of repeat prescriptions	2% ± 0.137	1% ± 0.120	1% ± 0.114	0.849
Video consultation	60% ± 0.492	64% ± 0.482	67% ± 0.470	0.190
Teleconsultation	12% ± 0.324	6% ± 0.235	9% ± 0.292	0.019 *
Telemonitoring	7% ± 0.258	4% ± 0.185	5% ± 0.228	0.113
Other	3% ± 0.167	2% ± 0.143	4% ± 0.200	0.179

One-way ANOVA was used to test differences in the mean use of specific e-health applications between the three practice types. * Indicates a significant difference (*p* < 0.05) between the mean values of the three practice types.

**Table 2 ijerph-18-12479-t002:** Percentage of practices using e-health, by e-health application (mean ± standard deviation), *n* = 735 (July 2020).

E-Health Application	Solo Practice (*n* = 149)	Duo Practice (*n* = 304)	Group Practice (*n* = 282)	*p*-Value
E-consultation	83% ± 0.381	83% ± 0.380	83% ± 0.373	0.964
Online ordering of repeat prescriptions	81% ± 0.392	87% ± 0.331	88% ± 0.330	0.134
Video consultation	44% ± 0.498	51% ± 0.501	56% ± 0.498	0.058
Teleconsultation	51% ± 0.502	67% ± 0.417	70% ± 0.460	0.000 *
Telemonitoring	8% ± 0.273	12% ± 0.324	13% ± 0.338	0.289
Other	7% ± 0.262	5% ± 0.217	10% ± 0.295	0.095
None	5% ± 0.212	1% ± 0.099	1% ± 0.118	0.019 *

One-way ANOVA was used to test differences in the mean use of specific e-health applications between the three practice types. * Indicates a significant difference (*p* < 0.05) between the mean values of the three practice types.

## Data Availability

The data presented in this study are available on request from the corresponding author.

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
