# Peer review of "Use of E-Health in Dutch General Practice during the COVID-19 Pandemic"

_ijerph, 2021, doi:10.3390/ijerph182312479_

Round 1

Reviewer 1 Report

This paper presents the study of the impact of the pandemic on the use of e-health in Dutch and the intention of practices to continue using e-health more intensively. The study is based on the use of web surveys sent to general practices in April and July 2020. Statistical analysis is performed and differences in the use of e-health between practice types are tested using ANOVA tests. The study shows that although there was an increased uptake of e-health in Dutch general practice during the COVID-19 pandemic, only a minority of practices indicated that they will maintain this increased use in the future.

The paper is well organized and clearly written. The methodology is correctly described too, although the study is somehow quite simple. Its main value is maybe the collection of many surveys and the discussion of them. However, as main concerns with the work one would have expected a more detailed and complete survey definition: a complete evaluation plan including more indicators regarding e.g. time duration of the consultations dedicated to every patient, staff organization changes, technical problems found during consultations, patient and staff satisfaction, etc. indeed a complete impact study of the new services provided from several perspectives (economical, organizational, acceptability, technical, access, patient perspective, etc.). There are some frameworks to evaluate eHealth services that could have been considered to improve the surveys and add more information to the questionnaires (some references follow this comment).

Bush M, Lederer AL, Li X, Palmisano J and Rao S. The alignment of information systems with organizational objectives and strategies in health care, International Journal of Medical Informatics, 2009, Vol 78 (7), pp. 446‐456.

Kidholm K, Ekeland AG, Jensen LK, Rasmussen J, Pedersen CD, Bowes A, Flottorp SA, Bech M. A model for assessment of telemedicine applications: MAST. Int J Technol Assess Health Care. 2012 Jan;28(1):44‐51.

European code of practice for telehealth services. 2014. Produced by the partners of the TeleSCoPE Project. www.telehealthcode.eu.

MOMENTUM. http://www.telemedicine‐momentum.eu

In summary, the study shows two main conclusions which are the core of the work: since the start of the pandemic, the use of e-health has rapidly increased and that the majority of general practices do not intend to continue their intensified use of e-health after the pandemic. However, no more significant research contribution other that this can be found in the paper, which might more suitable for a conference paper.

Reviewer 2 Report

  1. The introduction – The purpose of the study and messages aren’t delivered clearly. There is a lack of difference between this paper with other e-health papers. Lack of research significance in contribution.
  2. The paper should have a designated Literature Review part for e-health
  3. The theoretical framework is missing, the authors should clarify the reasoning behind the study.
  4. Hypotheses are also missing. I cannot follow what the authors are testing using ANOVA. It is not clear what are variables the authors are testing after reading part 3.4.
  5. If the authors are using ANOVA, they need to mention what test, they are doing (e.g., one-way, two-way, 2X2X2, etc.).
  6. Table 1 and Table 2 doesn’t represent the ANOVA results table. Please kindly check previous research on how they present their ANOVA results.
  7. The Descriptive Results as shown in Figures 1 to 3, kindly revise in a more professional way the tables.
  8. You mentioned “The Netherlands,” is 'the' – capital letter ‘T”
  9. As mentioned in the Discussion part, 2nd paragraph, “these findings are in line with the comparable results from different countries.” So what’s the difference between this study from others?
  10. There is no Practical and Research Implications
  11. Kindly improve your grammar writing…

Reviewer 3 Report

The manuscript by Keuper and colleagues aims to investigate the use of e-health in general practice during the initial phase of COVID-19 pandemic in the Netherlands. The topic is interesting and definitely deserves attention because pandemic resulted (and still does) in enormous challenges for the operation of health care.

The manuscript is generally well-written and I have some minor comments that, in my opinion, could improve its readability. Also, there are two more important issues that I would like you to consider with regards to study design and results' presentation (see comments 7 & 8).

  1. I suggest stating explicitly in the title that the study used data from the Netherlands and the initial phase of the pandemic. With these, the title would be more informative.
  2. I guess the first sentence in the introduction is an un-deleted part of the template. Please remove it.
  3. lines 51-52: it is not clear what the percentages given actually mean. After reading the next sentence, it seems that these are shares of practices using a particular type of e-health; however, this should be clearer earlier.
  4. lines 53-55: any specific data (concrete figures) could be provided that exemplifies the more developed e-health in the Netherlands compared to other European countries.
  5. line 67: what do you mean by 'almost all'? Please provide explicit figures. When writing about methodological aspects of research more rigor is needed.
  6. line 69: I suggest using 'NIVEL' (if this is something you mean) instead of ‘our’.
  7. An interesting insight into the variation of e-health use by the type of provider would be to account not only for the size of practice by also its ownership. I do not know if this is doable and applicable for Dutch healthcare. You do not provide detailed background on the organization of the GP system in the Netherlands (maybe all the practices are public or private). However, if there is a mix of private and public entities, and you have the information on the ownership in your registry, this could be very beneficial for your study to include this aspect into the analysis. Maybe private/public practices are more in favor of e-health use than the other. This could be very interesting.
  8. I guess the last two paragraphs in section '3.4. Practice type differences' lack tables. You describe the findings; however, I cannot see the figures behind these descriptions (correct me if I am wrong). Possibly you omitted these figures for the sake of brevity; however, your manuscript is quite short, so I think that readers could benefit from having a chance to see the figures you write about.

All and all, the manuscript is interesting, it describes well-designed and innovative research and, with some improvements made, will be a great paper for the journal.

Thank you for the opportunity to review it.

Reviewer 4 Report

It seems that the article would gain value if the authors showed their research in the light of theory, in the beginning, even in the Abstract. Thanks to this, not only practitioners but also theorists could be encouraged to analyze the article.

I encourage the authors to analyze the published results of research on Covid-19 as much as possible and not only from the "English-speaking research area", but more broadly from Central and Southern Europe, especially from those countries that were struggling with a pandemic, for example, Spain, Germany, Poland, Czech Republic, Italy. 
It is crucial because the exciting e-health service issues, which many countries have to deal with, has consequences. In light of the results of various studies, it also appears that the use of e-health service is caused by factors that seem distant from e-health yet strongly related, for example, skilfully managed framing or public trust in the State, its institutions, and trust in others.
After the good practical description of the approach to data collection, please add that literature studies were also used. I understand that also in this field, the research allowed for drawing specific conclusions. Please clearly determine whether your study confirmed or denied (to what extent) the results of other previous studies on e-health services. Please clearly define the research gap, research problems.
I believe (based on the review of 40 articles) that the rapid increase in e-medical services took place in "Western" countries and all over Europe. Examples are the Czech Republic, Poland, East Germany, Italy and Hungary. Please pay attention to this and expand the research field so that the article gains even more cognitive value.
Many English speaking researchers use a different source of information (written in other than English to collect necessary information during the study). I found, for example, that during the pandemic, a large percentage of Poles became convinced of all forms of remote medical services. According to a Biostat study, in 2017 only 6.8 per cent, Poles said that they had used telemedicine at least once in their life. And in 2020, as much as 60 per cent. Why? I do not know. It can be an interesting issue. Besides, one may find exciting data from German and Italian web pages too. I show this information to convince the authors to be careful in formulating the conclusion that "To our knowledge, this is the first study to investigate the impact of the COVID-19 pandemic on the (first time and intensified) use and intention to continue using e-health applications more intensively in general practice ". It seems better to say that, based on the published English-language literature, it can be concluded that ...;  Moreover, after analyzing the available published research, it can be concluded that the issues presented in this article are also analyzed from the perspective of other countries, including China (studies are published in English); Therefore,  I encourage the authors to systematically review the literature and draw appropriate conclusions only based on this review. Please carry out, if possible, a systematic literature review; please mention it in the methodology, and please show the result of the in-depth literature review. 

Reviewer 5 Report

E-health is not a relatively novel topic, but the authors have fully elaborated the latest trends and issues about this topic during the covid-19 pandemic, and made some certain interesting scientific contributions in the research area.
The research method is not only reasonable, but the authors fully realized the limitations of this study. In addition, the conclusion has been drawed through completely summarizing the implementation of experimental procedures.
The only problem is that authors have repeatedly mentioned the problem of "high cost", but has not yet provide enough data or references to support this view. (Such as "the costs may be performed too high to sustain their usage." on line 255.) I think the author needs to take this point into account in subsequent revisions.
In a word, beyond the minor problems highlighted above, the proposed work does not present significant weaknesses.

Round 2

Reviewer 1 Report

The new version of the paper still lacks of more detailed surveys or questionnaires to bring significant research value: indicators such as e.g. time duration of the consultations dedicated to every patient, staff organization changes, technical problems found during consultations, patient and staff satisfaction, etc. should be used in the questionnaires.

Although authors claim that extending the survey with additional questions about the use of eHealth would have resulted in lower response rates the study would have gained a lot in terms of research consistency. All the conclusions are based on almost a single question of use (yes/not), but without any other information related to how the consultations were carried out.

In summary, the paper has only undergone make-up changes from previous version.

Author Response

We thank the reviewer for commenting on the revised version of our manuscript. We do agree with the reviewer that - in retrospect - it would have been interesting to collect the additional data and indicators as mentioned (i.e. time duration of the consultations, staff organization changes, technical problems, patient and staff satisfaction). This would enable an impact study of the e-health services taking into account several perspectives (economical, organizational, acceptability, technical, access, patient perspective), which was also earlier mentioned by the reviewer.

However, we are limited to the survey data and indicators that could be collected at the time of the survey, being the start of the pandemic. At that time we were bound to use a minimum set of questions to execute our study. Given these limitations, we believe that the data collected provides relevant insights into e-health applications adopted and used by general practices during the initial phase of the COVID-19 pandemic, which we would like to share with the target audience of this journal. It goes without saying that it is not feasible to redo the survey retrospectively.

We subsequently suggest to extend our Discussion section on limitations and future research by addressing this important feedback of the reviewer, as we think that the type of study suggested deserves to be followed-up. 

Reviewer 4 Report

No comments 

Author Response

Thank you for your review.